# The Effect of Acetaminophen on Running Economy and Performance in Collegiate Distance Runners

**DOI:** 10.3390/ijerph19052927

**Published:** 2022-03-02

**Authors:** Riley P. Huffman, Gary P. Van Guilder

**Affiliations:** Department of Recreation, Exercise & Sport Science, Western Colorado University, Gunnison, CO 81230, USA; riley.huffman@western.edu

**Keywords:** endurance, time trial, perceived exertion, pain reliever

## Abstract

Acetaminophen (ACT) may decrease perception of pain during exercise, which could allow runners to improve running economy (RE) and performance. The aim of this study was to determine the effects of ACT on RE and 3 km time trial (TT) performance in collegiate distance runners. A randomized, double blind, crossover study was employed in which 11 track athletes (9M/2F; age: 18.8 ± 0.6 years; VO_2_ max: 60.6 ± 7.7 mL/kg/min) completed three intervention sessions. Participants ingested either nothing (baseline, BSL), three gelatin capsules (placebo, PLA), or three 500 mg ACT caplets (ACT). One hour after ingestion, participants completed a graded exercise test consisting of 4 × 5 min steady-state stages at ~55–75% of VO_2_ max followed by a 3 km TT. There was no influence of ACT on RE in any stage. Similarly, ACT did not favorably modify 3 km TT performance [mean ± SD: BSL = 613 ± 71 s; PLA = 617 ± 70 s; ACT = 618 ± 70 s; *p* = 0.076]. The results indicate that ACT does not improve RE or TT performance in collegiate runners at the 3 km distance. Those wanting to utilize ACT for performance must understand that ACT’s benefits have yet to be significant amongst well-trained runners. Future studies should examine the effects of ACT on well-trained runners over longer trial distances and under more controlled conditions with appropriate medical oversight.

## 1. Introduction

During high intensity efforts runners experience a great deal of pain [1]. This pain can be a result of muscle fatigue, tissue damage, or aggravation of previous injury [1]. In elite races, where all runners are well-trained with comparable aerobic capacities, pain management is often the primary factor for determining success [2]. Taking pain-relieving medications, which are not banned by the governing body of the sport, and which are safe to ingest for healthy individuals with no allergies to their ingredients or contraindications to the medication, has been investigated as a way to enhance exercise performance. Acetaminophen (ACT), also known as paracetamol, is an over-the-counter pain reliever and fever reducer. ACT can alter acute and chronic responses to exercise by increasing pain threshold and demanding a greater amount of a stimulus before pain is felt [3]. 

The use of analgesics is extremely prevalent among runners [4]. In a study of 806 runners conducted by Rosenbloom et al., researchers found that 87.8% of subjects had utilized analgesics within the last year [4]. Over 200 of the subjects in the study reported the use of non-steroidal anti-inflammatory drugs (NSAIDs) directly prior to a race event. The top three reasons for use before a race are: (1) to reduce inflammation/swelling (58%), (2) to increase pain tolerance (42.7%), and (3) to continue running through an injury (42.7%) [4]. As the use of NSAIDs and analgesics in running is highly prevalent for these three primary reasons, it is important for runners that wish to utilize these drugs to understand how the drug they choose to use acts on the body and the risks associated with each drug. 

ACT is considered to be a selective cyclooxygenase-2 (COX-2) inhibitor [5]. Being a selective COX-2 inhibitor, ACT lacks the antiplatelet and detrimental effects on gastrointestinal mucosa of COX-1 inhibition, making it safer on the gastrointestinal tract than other drugs that are not selective inhibitors. Most NSAIDs, such as ibuprofen, are not selective inhibitors and therefore contain a risk of intestinal damage. Unlike NSAIDs, ACT is almost unanimously considered to have no anti-inflammatory activity and does not produce gastrointestinal damage or untoward cardiorenal effects [6]. Indeed, a major adverse effect of NSAIDs is their known tendency to cause gastrointestinal (GI) complications, such as mucosal ulceration, bleeding, perforation, and the formation of diaphragm-like strictures [7]. In a study analyzing the aggravation of exercise-induced intestinal injury by ibuprofen in athletes, Van Wijck et al. examined four scenarios around ibuprofen and exercise (800 mg ibuprofen before cycling, cycling without ibuprofen, 800 mg ibuprofen at rest, and rest without ibuprofen intake). They found that ibuprofen consumption and cycling resulted in increased plasma intestinal fatty acid binding protein (I-FABP) levels, reflecting small intestinal injury. These levels were higher after cycling with ibuprofen than after cycling without ibuprofen, rest with ibuprofen, or rest without ibuprofen. Additionally, small intestinal permeability increased, especially after cycling with ibuprofen, reflecting loss of gut barrier integrity. They concluded that ibuprofen aggravates exercise-induced small intestinal injury and induces gut barrier dysfunction in healthy individuals [7]. This phenomenon does not occur with ACT due to the different mechanisms of action of ACT compared to ibuprofen and other NSAIDs [5]. These studies demonstrate that those wishing to utilize drugs prior to races in order to increase pain tolerance are at an increased risk of adverse health effects when choosing NSAIDs, such as ibuprofen, as opposed to an analgesic drug, such as ACT [7]. They also demonstrate that ACT should not be used to treat inflammation or swelling, as ACT does not have an anti-inflammatory effect [5,6]. 

ACT is a safe drug at appropriate doses [6]. The amount of 7.5 g in adults is widely considered as the lowest acute dose capable of causing toxicity [6]. All studies examining ACT and its effect on performance utilize doses ranging from 0.5 g to 1.5 g [3,8,9,10,11,12,13,14], well under the threshold for the potential of toxicity. There have been no reports of acute toxicity in healthy adults ingesting a single dose of ACT below 125 mg/kg [6]. Unlike ibuprofen or other NSAIDs, ACT has only a small peripheral effect and acts primarily on the central nervous system [6]. Even so, the risks of ACT should be fully outlined for coaches and athletes to consider. As with many drugs, ACT can have very harmful effects, specifically to the liver, if taken above prescribed doses. Unfortunately, ACT overdose is responsible for more acute liver failure cases in the US and UK than all other etiologies combined [15]. The most common reason ACT ingestion results in death by overdose is its use in suicide attempts. Suicide attempts are a frequent cause of exposure to a single, high overdose of ACT [15]. Regrettably, unintentional overdose can occur as a result of combining multiple over-the-counter drugs, such as sleep-aids and cold medications, that may all have components of ACT [15]. Nevertheless, extensive literature reviews suggest that even susceptible people are unlikely to suffer adverse effects from therapeutic doses of ACT [15]. Additionally, there is an antidote against ACT-induced liver injury, the drug, N-acetylcysteine (NAC). NAC acts through facilitating scavenging of a reactive metabolite during the metabolism phase and is most effective when administered within 8 h of the overdose. This allows ACT-induced liver injury and liver failure to have a relatively high survival rate. As ACT is one of the most common over-the-counter drugs and because the vast majority of those who take ACT do not come close to taking over-therapeutic dosages of it, it is one of the safest over-the-counter drugs available [15]. 

Several studies have demonstrated significant endurance performance improvements among participants in ACT conditions compared to placebo conditions during both cycling and running. These improvements have been contributed to improved ability to tolerate pain as a result of prolonged exercise or a decreased perception of perceived pain or exertion during exercise. For example, the cycling studies by Delextrat et al. [8], Foster et al. [9], Mauger et al. [10], Mauger et al. [11], and Morgan et al. [12] have demonstrated that ingestion of ACT before a cycling bout improved performance through an increased average or peak power output or a decreased time to complete a specified distance [8,9,10,11,12]. Some researchers concluded that these improvements in performance were a result of the participants’ improved ability to tolerate pain during cycling [8,9,10]. Despite the many studies completed on cycling performance and ACT, there have been few studies conducted to examine the effects of ACT ingestion on running performance. In the only study examining running endurance performance and ACT ingestion, Dagli et al. [3] found that after taking ACT, recreationally active runners were able to improve 3 km time trial performance by 1.9% compared to placebo [3]. However, this study was conducted using exclusively male participants and was performed on a treadmill, which is not as specific to distance running compared to the running over ground that is characteristic of National Collegiate Athletic Association (NCAA) track and cross-country runners. Indeed, there are no well-controlled randomized studies investigating the potential exercise performance effects of ACT on well-trained, elite level male and female distance runners. Additionally, there have not been any studies examining the effects of ACT on running economy (RE).

RE is determined by the steady state oxygen consumption for a standard speed [16,17,18]. An athlete with improved RE consumes less oxygen for a given steady state running speed [19], thus improving their performance by expending less energy throughout a race [19]. Over the course of a race, runners experience increasing amounts of fatigue, which contributes to reduced mechanical efficiency and poor economy of motion [1]. For instance, Meardon et al. [1] found that stride time became less consistent over the course of a 5 km time trial while examining stride time variability, indicating that during prolonged running there was an increased need for gate adjustments due to increasing fatigue [1]. 

Based on the evidence to date in a multitude of studies, ACT has been demonstrated to improve cycling performance [8,9,10,11,12]. Yet, the only study examining the effects of ACT ingestion on endurance running performance used recreationally active runners and did not examine its effect on RE [3]. Therefore, the purpose of this randomized, double blind, crossover experiment was to determine the effects of ACT on RE and a 3 km time trial performance in well-trained NCAA collegiate distance runners. It was hypothesized that ACT would improve RE and 3 km time trial performance through a reduction in perceived pain during running.

## 2. Materials and Methods

### 2.1. Experimental Approach

In this randomized, double blind, crossover experiment, participants reported to the High Altitude Exercise Physiology Laboratory on five separate occasions (see Figure 1). The experiment was randomized by order; the research assistant randomly assigned each participant’s first condition as either a baseline condition (BSL), only water ingestion, placebo condition (PLA), water and placebo ingestion, or ACT condition (ACT), water and ACT ingestion. The random assignment was accomplished using a random number generator, which selected a number at random between one and three. Participants proceeded to complete each session based on their starting condition. For example, if a participant was randomly assigned ACT as their first condition, their condition order would be ACT then BSL then PLA. The first session for all participants consisted of completing questionnaires, informed consent, and a treadmill running familiarization. This session lasted 30–45 min and was followed by the next session two days later. In the second session, lasting about an hour, anthropomorphic measurements and determination of VO_2_ max were completed. In the third, fourth, and fifth sessions, each occurring one week apart and commencing on the same day of the week at the same time of day, subjects were assigned to ingest either eight ounces of water to serve as their baseline (BSL), eight ounces of water paired with three empty red gelatin capsules which served as a placebo (PLA), or eight ounces of water paired with three 500 mg capsules of ACT (1.5 g) (ACT). Following ingestion, participants waited 60 min and then completed a 20-min RE assessment on a treadmill, which also served as a warm-up for their 3 km time trial on the indoor track.

### 2.2. Subjects

Eleven total participants (9 men and 2 women) were recruited to participate in the study. Participants were well-trained distance runners on the NCAA division II cross-country team at Western Colorado University (WCU). Participants were considered well-trained based on VO_2_ max and were in the 99th percentile for VO_2_ max in their age and sex-group based on fitness guidelines [20]. Participants completed the study during the middle of the indoor track season. They typically performed two workouts per week and four low intensity runs per week in addition to the experimental trials. One workout would consist of four to ten intervals of 400 to 1000 m with one to three minutes of jogging between repetitions. The other workout would be a continuous run of 20–35 min at 80–90% of VO_2_ max. Men in the study ran between 80 and 120 km/wk. Women in the study ran between 56 and 96 km/wk. Men and women runners trained at an average running velocity of 14.0 km/h and 12.5 km/h, respectively. Participants were excluded from the study if they were found to be allergic to, or had previous complications with, the drug ACT, if they were heavy alcohol users, or if they had had any liver complications in the past. Participants were also excluded from the study if they were not classified as low risk for heart disease based on the American College of Sports Medicine risk algorithm. Exclusion criteria were assessed with the physical activity readiness questionnaire (PAR-Q) [21] and a medical history questionnaire, which included questions regarding over-the-counter drug use and alcohol use. All measurements of participants were conducted in the High Altitude Performance Laboratory at WCU, except for the 3 km time trial, which was performed on the indoor track in the WCU Mountaineer Fieldhouse. All subjects provided written and verbal informed consent prior to participating in the study. This study was approved by the Institutional Review Board at WCU [HRC2020-01-01-R12].

### 2.3. Procedures

#### 2.3.1. Familiarization and Lead-In

Following completion of the informed consent and other screening questionnaires, participants underwent a lead-in period to familiarize them with the VO_2_ max protocol and treadmill RE assessments. The familiarization session was a way for participants to gain an understanding of how to run on a treadmill with open-circuit indirect calorimetry and to get a sense of the rating of the perceived exertion (RPE) scale. In this session the participant did not run to volitional exhaustion. This allowed them to return to the lab within 48 h for the genuine VO_2_ max assessment without the possibility of fatigue.

For this familiarization session participants were fitted with a mask attached to falconia tubing, which was attached to the metabolic cart (Parvo Medics TrueOne^®^ 2400, Sandy City, UT, USA) to collect expired gases. Participants were also fitted with a chest strap (Polar, Lake Success, NY, USA) to monitor heart rate throughout the test. Participants ran for 10 to 12 min at increasing speeds on the treadmill (Trackmaster, Newton, KS, USA). This session was paced in a way that participants would reach an RPE of about 7 within 10–12 min, giving them an understanding of the perception of effort they would feel throughout the genuine VO_2_ max assessment and later RE assessments. At the conclusion of the session, participants completed a low intensity cool-down at a self-selected pace for at least five minutes and were dismissed from the lab. The entirety of the screening and familiarization session lasted 30–45 min.

#### 2.3.2. Anthropomorphic Measurements

While wearing only running attire, the participant removed their shoes and stood on a scale (Tanita, Arlington Heights, IL, USA) to be weighed in kilograms (kg). A measuring stick built into the scale was used to measure the participant’s height in centimeters (cm). Body mass index (BMI) was later calculated from these measurements using the formula BMI = weight (kg)/height (m)^2^. Body fat percentage was assessed using an Omron HBF-300 handheld body composition analyzer (Omron, Bannockburn, IL, USA).

#### 2.3.3. Maximal Oxygen Consumption

Following anthropomorphic measurements, participants completed a self-selected 10-min dynamic warm-up consisting of stretches and other exercises on the indoor track in the WCU Mountaineer Fieldhouse. The same warm-up routine was performed for each participant prior to the RE test and time trial. This allowed the participant to prepare to perform as they would in a typical training session or race. 

VO_2_ max was determined using open-circuit spirometry combined with indirect calorimetry (Parvo Medics TrueOne^®^ 2400, Sandy City, UT, USA) in response to incremental treadmill running (Trackmaster, Newton, KS, USA). Flow and gas calibrations were performed prior to each test using standard operating procedures provided by the manufacturer. Participants were fitted with a mask attached to falconia tubing, which was attached to the metabolic cart to collect expired gases. They were also fitted with a chest strap (Polar, Lake Success, NY) to monitor heart rate throughout the test. The treadmill was set to an initial incline of one percent grade, as one percent grade most accurately reflects the energetic cost of outdoor running [22,23]. 

Male participants completed a 3-min warm up at 12 km/h at a 1% grade. Thereafter, the treadmill velocity was increased 0.8 km/h every minute until velocity reached 19.2 km/h; at this point, the velocity remained constant and grade of the treadmill was increased 2% every minute of the test until the participant reached volitional fatigue. Female participants followed a similar pattern. They completed a 3-min warm up at 10.5 km/h at a 1% grade. Thereafter, treadmill velocity was increased 0.8 km/h every minute until velocity reached 17.7 km/h. At this point the velocity remained constant and grade of the treadmill was increased 2% every minute of the test until the participant reached volitional fatigue. Heart rate and RPE were recorded at the end of each minute throughout the test. Participants were provided verbal encouragement throughout the test until exhaustion. VO_2_ data were smoothed with a 15-s moving average. VO_2_ max was denoted as the highest 15 s moving average obtained during the last minute of exercise with no further increase in VO_2_. All tests were terminated by volitional exhaustion. A true VO_2_ max was confirmed based on a plateau in VO_2_ defined by a change of <150 mL/min despite a change in workload and an RER greater than 1.10.

#### 2.3.4. Intervention Sessions

Each of the three intervention sessions took place on the same day of the week, beginning at the same time of day, exactly one week apart for each participant. Participants were instructed to avoid caffeine for four hours prior to all interventions and tests. They were also instructed to maintain similar diet and sleep habits on the days prior to testing and on the days of testing. Each session began with the participant meeting with the research assistant who would administer one of the three interventions. The participant received either a baseline of 8 ounces of water paired with nothing, three placebo capsules paired with 8 ounces of water, or 1.5 g ACT in the form of three 500 mg capsules paired with 8 ounces of water. The order of the intervention was randomized and was blinded to the participant and the primary investigators. Following ingestion, the participant was instructed to relax and perform a non-stressful activity, such as reading or listening to music for 50 min. Thereafter, participants completed their individualized 10-min dynamic warm up routine, and then transitioned to the RE test. This timeframe was chosen because peak plasma concentration of ACT occurs approximately 45–60 min after oral administration [6].

#### 2.3.5. Running Economy 

Each participant was fitted with a heart rate monitor and a mask connected to a tube leading to the metabolic cart in the same fashion as the familiarization and VO_2_ max tests. The RE test consisted of four and five-minute stages at increasing running velocities. The intensity of each stage was kept relatively low as RE at lower speeds has been demonstrated to be more strongly correlated with performance [16]. The duration of five minutes for each stage was selected, as it takes approximately four to five minutes to reach steady state oxygen consumption [24]. The total duration of 20 min was selected because the participants in the study consistently warm-up for workouts and race for 20 min and the RE portion of the assessment served as a warm-up for the subsequent 3 km time trial performance measure. Heart rate, oxygen consumption, and RPE were recorded during the last minute of each stage. Male participants ran each stage at 10.5 km/h, 11.2 km/h, 12.0 km/h, and 12.9 km/h (174.4 m/min, 187.8 m/min, 201.2 m/min, and 214.6 m/min). Female participants ran each stage at 9.7 km/h, 10.5 km/h, 11.2 km/h, and 12.0 km/h (160.9 m/min, 174.4 m/min, 187.8 m/min, and 201.2 m/min). The intensity of these stages ranged from approximately 55% to 75% of VO_2_ max throughout the duration of the test. RE was expressed in two ways. First, as the oxygen cost required to run 1 km of horizontal distance (mL/kg/km) and second, as the caloric unit cost—the energy in kilocalories required to run 1 km of horizontal distance (kcal/kg/km). This unit has been demonstrated to be more sensitive to changes in relative velocity compared with oxygen cost [25]. Caloric unit cost was calculated by dividing the steady-state energy expenditure cost (kcal/min) obtained during the last four 15-s moving averages of each stage by body mass (kg), divided by running velocity (m/min) and multiplied by 1000 (1000 m/km), as done in a previous RE study [24].

#### 2.3.6. Track Time Trial

Following the RE test, participants were given 15 minutes to use the restroom, change shoes if desired, and perform any additional warm up stretches or exercises as they normally would prior to a race. This routine was again kept constant for each individual participant. Exactly 15 min following the conclusion of the RE test, the participants began the 3 km time trial on the indoor track at the Mountaineer Fieldhouse. 

Participants were instructed to complete fifteen laps on the 200 m track in the innermost lane as quickly as possible, as they would in a race. Researchers recorded the duration of each lap split. At the completion of the time trial, the total run duration, heart rate, and oxygen saturation were recorded. The participant’s heart rate and oxygen saturation (SpO_2_) were measured by a fingertip pulse oximeter (American Diagnostic Corporation, Hauppauge, NY, USA). The participant then performed a low intensity cool down of their choosing and subsequently completed a four-question survey regarding their perception of effort and difficulty during the time trial. The questions on the survey were answered on a 1–10 scale with 1 being lowest and 10 highest. The questions were: (1) How would you rate your level of effort over the course of the time trial? (2) How would you rate the exercise difficulty of the entire time trial? (3) How would you rate your level of performance based on your current fitness level during the time trial? (4) How would you rate your average level of perceived exertion over the time trial? Following the completion of the survey, the participant was dismissed for the day. Each remaining intervention was separated by exactly one week and repeated at the same time of day in the randomly determined order.

#### 2.3.7. Statistical Analysis

Descriptive characteristics of the participants are presented as means and standard deviations. With the exception of the 3 km time trial, all data met assumptions of normality. Therefore, the nonparametric Friedman’s test was used to determine treatment differences in the 3 km time trial, and data are reported with median and interquartile range as appropriate. A 3 × 4 (trial × exercise stage) analysis of variance for repeated measures with Bonferroni adjustment for pairwise multiple comparisons was used for treatment differences in submaximal exercise variables (i.e., VO_2_ and heart rate) and RE between the four treadmill stages. A three-way analysis of variance with repeated measures was used to determine differences in post time trial heart rate and oxygen saturation. A Friedman’s test was used to determine differences in RPE and the four-question self-evaluation form. These data were therefore displayed accordingly as median and interquartile range. All other continuous data that met assumptions of normality were reported as mean ± standard deviation. Level of statistical significance was set at *p* < 0.05 and SPSS version 27 (IBM-SPSS, Boston, MA, USA) was used to perform these statistical analyses.

## 3. Results

Table 1 shows subject characteristics. Runners were normal weight based on body mass index and presented with VO_2_ max values at or above the 99th percentile for age and sex [20]. Each participant continued normal track training during the study: the average mileage for the men was 103 km/wk; the average mileage for the women was 76 km/wk.

### 3.1. Time and Splits

Figure 2 shows the 3 km time trial performance for the interventions. There were no significant differences in the time to complete the time trial among the interventions. Performance times observed in the present study were within 2–5% of what the participants actually performed in a competitive 3 km race in the weeks following the study. Mean performance time between the three conditions were within five s (*p* = 0.076). Similarly, no significant differences were found in any of the 1 km split times between BSL, PLA, and ACT groups (*p* = 0.406, 0.234, and 0.811, respectively). Mean values for the first, second, and third split times from start to 1 km, 1 km to 2 km, and 2 km to 3 km were all within three seconds between BSL, PLA, and ACT conditions.

### 3.2. Running Economy

The RE exercise intensities for stage 1, 2, 3, and 4 corresponded to approximately 60%, 66%, 71%, and 74% of VO_2_ max, respectively. All subjects achieved steady-state oxygen consumption in each stage. As shown in Figure 3, there were no significant differences between BSL, PLA, and ACT conditions in RE expressed as mL/kg/km in any of the four stages. For example, the average RE across stage 4 for BSL was 207.4 mL/kg/km, compared with 208.6 mL/kg/km for PLA and 208.5 mL/kg/km for ACT (*p* = 0.886; Figure 3). Likewise, when expressed as kcal/kg/km, RE was also similar among the conditions. For example, the average RE across stage 4 for BSL was 1.009 kcal/kg/km, compared with 1.015 kcal/kg/km for PLA and 1.016 kcal/kg/km for ACT (*p* = 0.857; Figure 4).

### 3.3. Heart Rate, Oxygen Consumption, and Saturation

Table 2 shows the sub-maximal heart rate and oxygen consumption during the RE tests, and the post time trial oxygen saturation and heart rate. Sub-maximal oxygen consumption was similar between BSL, PLA, and ACT at each of the four RE stages (*p* = 0.529, 0.148, 0.234, and 0.159, respectively) as was heart rate (bpm) throughout the four RE stages (*p* = 0.518, 0.135, 0.071, and 0.176, respectively). Post time trial oxygen saturation (*p* = 0.913) and post time trial heart rate (*p* = 0.846) were not different between BSL, PLA, and ACT conditions.

### 3.4. RPE and Questionnaire Responses

As shown in Table 3, there were no significant differences in RPE during each RE stage between the three conditions. In addition, no significant differences were found in any of the post 3 km time trial questionnaire responses between the three conditions.

## 4. Discussion

In contrast to our hypothesis, the primary findings of this randomized, double-blind crossover trial indicate that supplementation with ACT does not improve RE or 3 km time trial performance in NCAA competitive cross-country athletes. Baseline RE expressed as mL/kg/km and kcal/kg/km were within 1% of the placebo and ACT conditions for all stages of the incremental exercise test. Additionally, there was no noticeable difference in the time to complete the 3 km time trial, as well as with 1 km splits with ACT. Collectively, given similar steady-state oxygen consumption, heart rate, minute ventilation, respiratory exchange ratio, and gross energy cost during incremental treadmill running between conditions, ACT supplementation does not favorably modify exercise economy or 3 km time trial performance.

ACT’s primary mechanism of action for pain relief is on the serotonergic descending pain pathway [26]. ACT inhibits prostaglandin (PG) synthesis from arachidonic acid in human skeletal muscle; as a result, PGs regulate adaptations to muscular exercise [26]. An analgesic drug, such as ACT, can alter the acute and chronic responses to exercise by elevating the pain threshold and requiring a greater amount of pain before it is felt [3]. As exercise-induced pain is a contributor to volitional exhaustion or changes in pacing during exercise [2], a reduction in perceived pain should increase the athlete’s performance [2,3]. 

RE has been demonstrated to account for a significant amount of the variation observed in race performance amongst elite level runners [16]. RE is dependent on a variety of factors, including efficiency of form [27] and both central and peripheral fatigue [1,28,29]. Due to the influence of RE to be affected by central fatigue and its correlation with performance in elite runners, it was hypothesized that ACT, which has a direct effect on central fatigue [26,30], would improve RE in competitive athletes. The results of this study demonstrate that RE was similar across BSL, PLA, and ACT conditions for all RE stages, both as measured in mL/kg/km and in kcal/kg/km.

Our findings are in contrast to those by Dagli et al. [3], who reported a 1.9% (14 s) improvement in the 3 km time trial performance. It is important to note the crucial study design and sample differences between the present study and those by Dagli and colleagues. While Dagli et al. studied recreationally active male runners with an average VO_2_ max of 55.67 ± 5.35 mL/kg/min, measured at sea level [3], we specifically recruited in-competition NCAA elite male and female athletes with an average VO_2_ max of 60.6 ± 7.7 mL/kg/min, measured at 7700 feet, which had already undergone substantial endurance training. Indeed, the participants in our study completed the 3 km distance two minutes faster on average than participants in the study by Dagli et al. [3], despite completing the trials at 7700 feet as opposed to sea level. Notably, the average post exercise oxygen saturation levels in the present study are quite low, as shown in Table 2. This difference in completion time and duration of effort may be a critical reason as to why the present study did not demonstrate appreciable differences in performance with ACT. 

Of the previous studies examining endurance performance, the majority of studies that found statistically significant differences in performance between ACT and control conditions required exercise times of at least 20 min. For example, Mauger et al. [10] found a 30 s improvement between ACT and control in a 10-mile cycle time trial, which lasted on average 26.25 min for the ACT condition and 26.75 min for the control condition [10]. Mauger et al. [11] found a four-minute difference between ACT and control conditions in a cycle test to exhaustion in the heat in which subjects in the ACT condition were able to cycle for 22.7 min compared to only 18.8 min in the control condition [11]. Foster et al. found a 19 watt improvement in average power between ACT and control conditions in a repeated Wingate study with a total exercise time of 20 min when active recovery cycling was included [9]. Delextrat et al. found a 24-watt improvement in peak power [8] in a similar study design as Foster et al. [9]. All of these studies used a randomized double-blind crossover design [8,9,10,11]. Only two previously conducted studies investigated the effects of ACT on exercise bouts involving an exercise duration less than 20 min [3,12]. Both of these studies were also randomized double blind crossover studies with an ACT condition and a placebo condition. These two studies reported significant performance benefits of ACT compared with control conditions. However, it should be emphasized that both studies demonstrated marginal, albeit significant, improvements with ACT. First, the study by Dagli et al. [3] reported a small difference of 1.9% with ACT. Second, the study by Morgan et al. [12] also demonstrated a small, 5-watt difference in average power over a 3-min maximum cycle test, which was a 1.4% difference compared with control conditions [12]. 

ACT has been demonstrated to decrease participants’ RPE during running bouts of similar or increased intensities [3,13]. We did not demonstrate any significant differences in RPE between treatment conditions. Likewise, we did not demonstrate any significant differences in the perception of effort and difficulty questionnaire among trials. The median reported value for all four of the post time trial questions on the questionnaire were within 1 point of each other on the 1–10 scale across the three treatment conditions. This displays that the participants did not perceive a significant difference in effort, difficulty, performance, or exertion between the three conditions. 

A limitation of this study was the small sample size (11) and unequal distribution of male (9) and female (2) participants. Although all statistical tests were performed while examining only one sex at a time and results were the same as when all participants were examined together, a more balanced quantity of male and female participants is recommended for future studies. Another limitation of the present study was the training schedule of participants was not controlled from week to week. Although participants completed similar weekly training volumes throughout the study, the exact workouts of the participants varied from week to week and thus may have resulted in the participants feeling more or less fatigued from each week’s workouts between the three trials. However, the variability between training loads should not have had a significant effect on the total outcome of the study, due to the randomization of the order each participant undertook. Future studies can minimize the potential for this phenomenon by scheduling data collection during the off-season, where the participants are less likely to be doing high intensity training over the study duration. Furthermore, the duration between trials may not need to be an entire week, as well-trained runners typically have the ability to recover from hard efforts faster than recreationally active runners. For example, it has been suggested that “aerobic fitness enhances recovery from high intensity intermittent exercise through enhanced aerobic contribution, increased post-exercise VO_2_, and possibly by increased lactate removal and increased PCr restoration, which has been linked to improved power recovery” [31,32]. Therefore, in studies involving well-trained participants, the required time between trials may only be 48–72 h as opposed to studies with recreationally active participants who require more time between trials to fully recover. 

Another variable that future research studies involving well-trained participants should examine is the trial distance. The present study examined the effectiveness of ACT over a relatively short time trial distance of 3 km. On average, participants completed this distance in just over ten minutes. ACT has been demonstrated to be effective in improving performance amongst well-trained participants over longer durations. For example, Mauger and colleagues examined ACT’s influence on performance over a ten-mile cycle time trial in 13 trained male cyclists [10]. They found that ACT significantly improved performance by 30 s over the course of the ten-mile trial. As this study was examining a distance that took over 26 min to complete, it is possible that ACT’s effectiveness increases as the required trial duration increases, especially in studies involving well-trained participants. Future studies on well-trained runners should investigate the effectiveness of ACT over distances of at least 8 km, which would require an average duration of effort of at least 26 min, as in the study by Mauger et al. [10].

## 5. Conclusions

The results of the present study indicate no performance benefit of ACT on RE or a 3 km time trial performance among NCAA competitive collegiate distance runners. These results are in contrast to some [3,8,9,10,11,12], but not all [13,14] studies investigating the potential for ACT to improve exercise performance. To our knowledge, as this is the first study to examine the effects of ACT on well-trained collegiate distance runners, it appears that ACT has a more significant effect on performance for recreationally active runners than well-trained runners [3]. It remains to be found whether ACT is beneficial for well-trained runners at distances other than 3 km or under more controlled experimental circumstances, such as with a higher number of participants and a more favorable proportion of male and female participants. Finally, it should be emphasized that ingestion of ACT is for medical purposes only. It is not recommended for athletes to use ACT for exercise performance without a medical indication. Moreover, prior review of relevant medical history by the athletes’ healthcare team, and physician approval before an athlete uses ACT, is the best practice.

## Figures and Tables

**Figure 1 ijerph-19-02927-f001:**
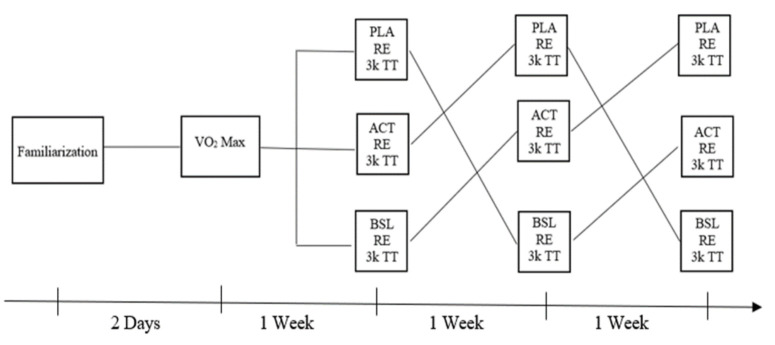
Experimental flowchart. (BSL = baseline) (PLA = placebo) (ACT = acetaminophen) (RE = running economy) (TT = Time Trial).

**Figure 2 ijerph-19-02927-f002:**
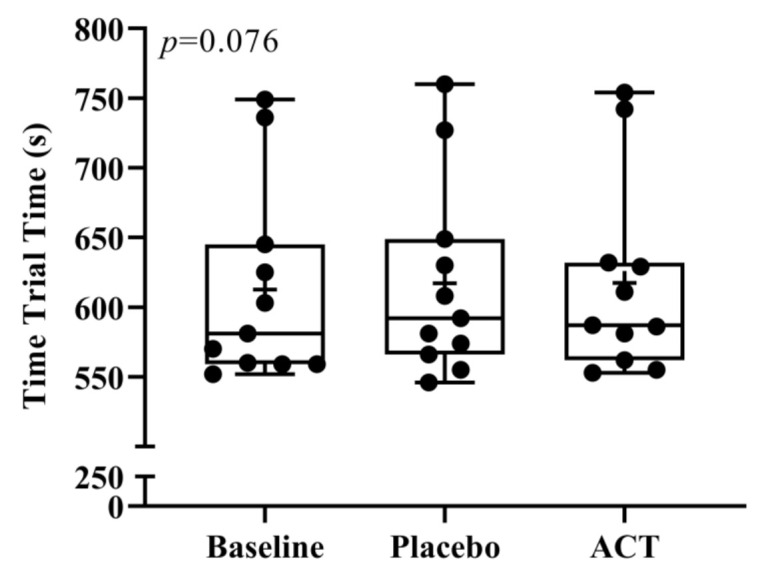
Box and whisker plot for 3 km time trial times in s for baseline, placebo, and ACT conditions. The box is the interquartile range, the horizontal line is the median, the cross (✛) is the mean, and the error bars are the minimum and maximum. ACT; acetaminophen.

**Figure 3 ijerph-19-02927-f003:**
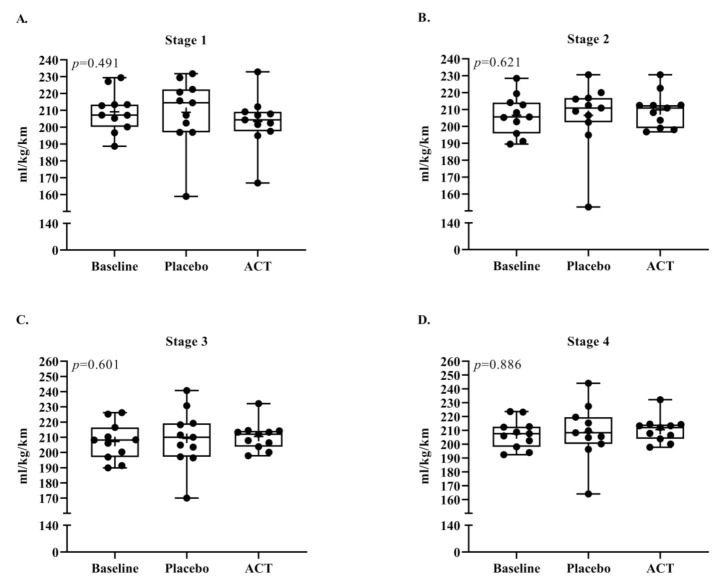
Box and whisker plots for RE expressed as mL/kg/km for baseline, placebo, and ACT conditions for (**A**) Stage 1: 60% of VO_2_ max, (**B**) Stage 2: 66% of VO_2_ max, (**C**) Stage 3: 71% of VO_2_ max, and (**D**) 74% of VO_2_ max. The box is the interquartile range, the horizontal line is the median, the cross (✛) is the mean, and the error bars are the minimum and maximum. ACT; acetaminophen.

**Figure 4 ijerph-19-02927-f004:**
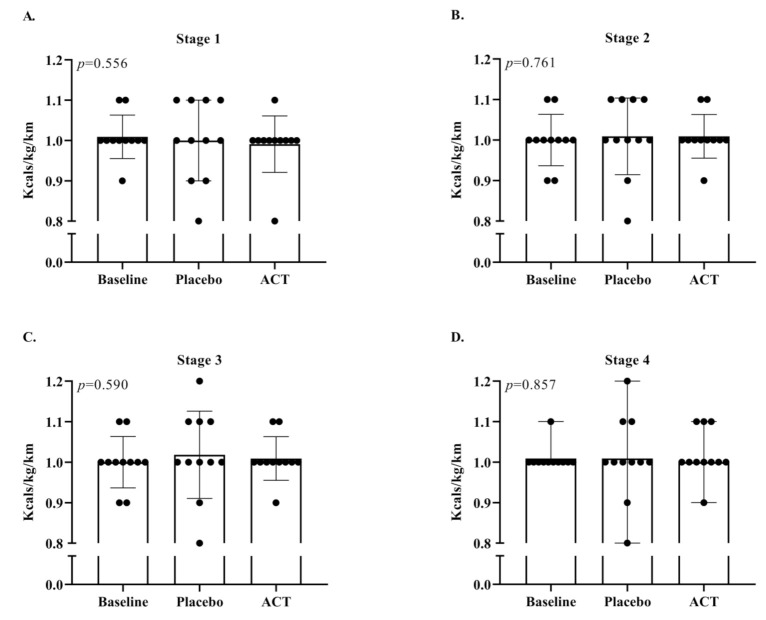
RE expressed as kcal/kg/km for baseline, placebo, and ACT conditions for (**A**) Stage 1: 60% of VO_2_ max, (**B**) Stage 2: 66% of VO_2_ max, (**C**) Stage 3: 71% of VO_2_ max, and (**D**) 74% of VO_2_ max. ACT; acetaminophen.

**Table 1 ijerph-19-02927-t001:** Descriptive characteristics of participants.

Characteristic (Units)	All Participants (*n* = 11)	Men (*n* = 9)	Women (*n* = 2)
Age (years)	18.8 ± 0.6	18.7 ± 0.5	19.5 ± 0.7
Height (cm)	171.1 ± 6.9	173.7 ± 4.2	159.5 ± 2.1
Weight (kg)	58.6 ± 5.3	59.7 ± 5.1	53.8 ± 2.9
BMI (kg/m^2^)	20.0 ± 1.1	19.8 ± 0.8	21.2 ± 1.7
BF%	9.7 ± 5.1	7.6 ± 1.0	19.5 ± 4.2
ABS VO_2_ max (L/min)	3.6 ± 0.6	3.8 ± 0.4	2.6 ± 0.3
REL VO_2_ max (mL/kg/min)	60.6 ± 7.7	63.3 ± 5.3	48.4 ± 2.2
Weekly training distance (km/wk)	98.1 ± 16.7	103.0 ± 9.9	76.0 ± 28.3

Abbreviations: cm (centimeters), kg (kilograms), kg/m^2^ (kilogram/meter^2^), BF% (body fat %) ABS (absolute), L/min (liters/minute), REL (relative), mL/kg/min (milliliters/kilogram/minute), and km/wk (kilometers per week).

**Table 2 ijerph-19-02927-t002:** Oxygen consumption and heart rate between baseline (BSL), placebo (PLA), and acetaminophen (ACT) conditions.

Stage	Speed (m/min) (Male, Female)	BSL (M ± SD)	PLA (M ± SD)	ACT (M ± SD)	*p*-Value
**Oxygen Consumption (mL/kg/min)**					
**1**	174.4, 160.9	36.0 ± 2.6	35.9 ± 3.6	35.0 ± 2.8	0.529
**2**	187.8, 174.4	38.3 ± 2.8	38.3 ± 3.7	38.9 ± 2.4	0.148
**3**	201.2, 187.8	41.2 ± 3.5	41.6 ± 3.5	41.8 ± 2.2	0.234
**4**	214.6, 201.2	44.0 ± 2.7	44.2 ± 3.9	44.2 ± 2.5	0.159
**Heart Rate (bpm)**					
**1**	174.4, 160.9	134 ± 11	139 ± 8	136 ± 9	0.518
**2**	187.8, 174.4	143 ± 10	150 ± 9	147 ± 9	0.135
**3**	201.2, 187.8	149 ± 8	155 ± 8	154 ± 8	0.071
**4**	214.6, 201.2	158 ± 8	162 ± 9	160 ± 8	0.176
**Post TT Oxygen Saturation (%)**					
N/A	(Post 3 km TT)	87.6 ± 2.8	87.6 ± 3.0	87.4 ± 3.1	0.913
**Post TT Heart Rate (bpm)**					
N/A	(Post 3 km TT)	176.7 ± 6.4	174.1 ± 11.3	175.7 ± 6.2	0.846

M ± SD, (mean ± standard deviation); mL/kg/min, (milliliters/kilogram/minute); m/min, (meters/minute); bpm, (beats per minute); TT, (time trial); km, (kilometer); BSL, baseline; PLA, placebo; ACT, acetaminophen.

**Table 3 ijerph-19-02927-t003:** Rating of perceived exertion (RPE) (1 = very low RPE, 10 = very high RPE) during RE stages and post TT perception of effort and difficulty responses between BSL, PLA, and ACT trials.

Stage/Question	Speed (m/min) (Male, Female)	BSL Med (25th, 75th)	PLA Med (25th, 75th)	ACT Med (25th, 75th)	*p*-Value
**RPE during RE Stages**					
**1**	174.4, 160.9	1 (1, 2)	1 (1, 2)	1 (1, 2)	0.273
**2**	187.8, 174.4	2 (1, 2.5)	2 (2, 2.5)	2 (1, 3)	0.957
**3**	201.2, 187.8	2.5 (1.5, 3)	3 (2, 3)	2 (2, 4)	0.519
**4**	214.6, 201.2	3 (1.5, 5)	3 (2, 5)	2 (2, 5)	0.922
**Responses to Questionnaire**					
**Q1**	Effort level	9 (8, 9)	9 (8, 9.5)	9 (9, 9)	0.091
**Q2**	Exercise difficulty	8.5 (8, 9)	9 (8, 10)	9 (8, 9)	0.656
**Q3**	Performance level	8 (8, 9)	8 (7, 9)	8 (7.5, 9)	0.368
**Q4**	Perceived exertion	8 (7, 9)	8 (8, 9)	9 (8, 9)	0.140

Abbreviations: TT (time trial); BSL, baseline; PLA, placebo; ACT, acetaminophen; Med 25th, 75th (median (25th percentile, 75th percentile)); m/min (meters/minute); Q, (question).

## Data Availability

Some or all data and models that support the findings of this study are available from the corresponding author upon reasonable request.

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
