# Peer review of "The Effect of Acetaminophen on Running Economy and Performance in Collegiate Distance Runners"

_ijerph, 2022, doi:10.3390/ijerph19052927_

Round 1
Reviewer 1 Report
Dear authors,
Thank you to give the opportunity to review your manuscript.
In general, is interesting to continue investigating about the possible benefits of the effect of acetaminophen, an interesting your results because are opposite from other authors. One the limitation in that study a similar are the small number of participants, it should be included on the limitation of the study.
Introduction
Line 38 are two points after running.
In general, some references are in APA, but this is not the citation style of IJERPH, review the introduction.
Line 50, NCAA is not defined before on the introduction.
Material and Methods
Line 78, (BSL, PLA, or ACT) should be defined to help the reader to understand better that point.
Subjects
Line 118 (PAR-Q) a reference is needed.
Discussion
One limitation is the small number of participating and the differences between men (9) and women (2) it can affect in the results an should be included on the discussion as a limitation.
Conclusions
One recommendation should be that other researcher repeat the study with a large sample and a higher proportion of women.
References
4, 7, 11, 13, 15, 16, 17, 20, 22. Are quite old, should be considerer to include some recent ones.
https://www.mdpi.com/2075-4663/9/9/126
https://journals.physiology.org/doi/abs/10.1152/japplphysiol.00108.2021
https://onlinelibrary.wiley.com/doi/full/10.1111/acem.14169?casa_token=JcAg3wT_Kq0AAAAA%3AiB-TFw11kj4zVZA5naMyxNvAkTolXigck20sY8Y5q_53TF4wudlMxT2OJBI002X-go099HRUrFJaNaJB
Best regards,
Reviewer 2 Report
The manuscript entitled “The Effect of Acetaminophen on Running Economy and Performance in Collegiate Distance Runners” (ijerph-1591825) is well and interestingly written, and presents well designed protocol and valuable data. However, it does contain very serious problems that need to be corrected before the manuscript can be reconsidered for publication.
General remarks
- The most serious problem is that the authors (possibly in good faith) are totally unacceptable and irresponsible in recommending the use of ACT. In its current form, it completely excludes the possibility of disseminating and publishing this work.
ACT are not "carbohydrate candies" or "jelly beans", but drugs for a specific purpose (they are not intended to support physical performance). You should not automatically recommend taking this type of measures if there is no pain - it is absolutely unacceptable to recommend this type of action. It cannot and should be forbidden to automatically recommend "efficiency" of taking this type of measures "just in case" if there is no pain - it is absolutely unacceptable to recommend this type of action.
- Conclusions and a significant part of the work must be thoroughly revised in this regard. I would like to point out that the research is interesting and correct - it is justified. The results are also valuable. But the authors should approach the subject rationally and take the responsibility - because their work can be used by athletes and coaches. The subject is in fact very controversial because it is difficult to indicate even in the subtext that the use of this measure is something normal for Running Economy and Performance - it would not be good if athletes would use these measures without a specific medical justification etc ... they should very clearly indicate and indicate the dangers and contraindications.
- It is strongly recommended to Include a thorough overview of the dangers and side effects as well as potential synergism / antagonism with other agents.
- The authors at work sometimes generalize "pain during exercise" - there are different types / sources of pain and it would be good to be more explicit.
- In the title, abstract and work, however, it is worth specifying clearly what type of run / distance this study concerns (and is limited to). From the title, it can be subconsciously understood that distance runners concern long runs, where the influence of ACT can be expected more.
- Conclusions in the abstract and manuscript should be limited only to what has been researched and proven. Currently, there are many extrapolations that are illegitimate and more hypothesized - there are places for that in the discussion section.
- Lines 27-28 - Again, the authors write: "Taking pain-relieving medications, which are safe and not banned by the governing" - again a bit not fully matured - safe, but under certain conditions, doses, in healthy people etc. - there are many dangers, especially in longer use / higher doses etc.
- Lines 37-38 – the authors wrote: “ACT has been shown to enhance performance in endurance events including cycling and running” - This is a simplification - after all, not directly, but indirectly, always, or only at the time of the onset of painful symptoms.
- Line 88 – Was not "eight ounces of water" also given before the PLA?
- Table 1 should be supplemented with training experience and training data, such as distance.
- Table 1 - Is male body mass normal? A bit abnormally low for adult males.
- Table 2 - The saturation result is very low - is this a reliable result? This should be discussed.
- Work edition: standardize the edition, standardize the citation notation (once in square brackets, once in round), correct superscripts / superscripts where applicable (eg “VO2max”, “-1”, “m2”), remove double dots. Furthermore, terms like "caloric cost" should be corrected to "energy cost".
Round 2
Reviewer 1 Report
Dear author,
Changes suggested have been applied.
Best regards,
Reviewer 2 Report
All comments have been addressed.